# Diverse deep-sea anglerfishes share a genetically reduced luminous symbiont that is acquired from the environment

Lydia J Baker[1]*, Lindsay L Freed[2], Cole G Easson[2,3], Jose V Lopez[2], Danté Fenolio[4], Tracey T Sutton[2], Spencer V Nyholm[5], Tory A Hendry[1]*

[1]Department of Microbiology, Cornell University, New York, United States; [2]Halmos College of Natural Sciences and Oceanography, Nova Southeastern University, Fort Lauderdale, United States; [3]Department of Biology, Middle Tennessee State University, Murfreesboro, United States; [4]Center for Conservation and Research, San Antonio Zoo, San Antonio, United States; [5]Department of Molecular and Cell Biology, University of Connecticut, Storrs, United States

**Abstract** Deep-sea anglerfishes are relatively abundant and diverse, but their luminescent bacterial symbionts remain enigmatic. The genomes of two symbiont species have qualities common to vertically transmitted, host-dependent bacteria. However, a number of traits suggest that these symbionts may be environmentally acquired. To determine how anglerfish symbionts are transmitted, we analyzed bacteria-host codivergence across six diverse anglerfish genera. Most of the anglerfish species surveyed shared a common species of symbiont. Only one other symbiont species was found, which had a specific relationship with one anglerfish species, *Cryptopsaras couesii*. Host and symbiont phylogenies lacked congruence, and there was no statistical support for codivergence broadly. We also recovered symbiont-specific gene sequences from water collected near hosts, suggesting environmental persistence of symbionts. Based on these results we conclude that diverse anglerfishes share symbionts that are acquired from the environment, and that these bacteria have undergone extreme genome reduction although they are not vertically transmitted.

DOI: https://doi.org/10.7554/eLife.47606.001

**\*For correspondence:**
lb693@cornell.edu (LJB);
th572@cornell.edu (TAH)

**Competing interests:** The authors declare that no competing interests exist.

## Introduction

Symbiosis between animals and bacteria can enable both organisms to adapt to harsh environments or expand into new habitats, which impacts the ecology and evolution of both bacterial and host lineages (*McFall-Ngai et al., 2013*; *Moran, 2007*; *Moya et al., 2008*). Symbiosis with luminescent bacteria has evolved independently multiple times in diverse squid and fish species (*Davis et al., 2016*; *Dunlap and Urbanczyk, 2013*) and has been correlated with host diversification (*Davis et al., 2016*; *Ellis and Oakley, 2016*). Bioluminescence is considered an adaptive phenotype across multiple taxa and a ubiquitous function in the largest habitat on the planet, the bathypelagic biome (ocean's mid-waters below 1000 m) (*Martini and Haddock, 2017*). Four genera of bacteria in the family Vibrionaceae engage in luminescent symbiosis (*Dunlap and Urbanczyk, 2013*; *Hendry et al., 2018*; *Schaechter, 2009*), including the model species *Aliivibrio fischeri*, but comparatively little is known about the bioluminescent symbionts of deep-sea anglerfishes. Ceratioid anglerfishes (suborder Ceratioidei) consist of 167 species from 11 families (*Froese and Pauly, 2018*) and are the most speciose fish suborder in the bathypelagic zone (*Pietsch, 2009*). Most female ceratioid anglerfishes host extracellular luminous symbiotic bacteria in a lure-like projection (esca) above the animal's head (*Munk, 1999*). The genera *Cryptopsaras* and *Ceratias* harbor bacterial symbionts in additional

**eLife digest** The deep sea is home to many different species of anglerfish, a group of animals in which females often display a dangling lure on the top of their heads. This organ shelters bacteria that make light, a partnership (known as symbiosis) that benefits both parties. The bacteria get a safe environment in which to grow, while the animal may use the light to confuse predators as well as attract prey and mates.

The genetic information of these bacteria has changed since they became associated with their host. Their genomes have become smaller and more specialized, limiting their ability to survive outside of the fish. This phenomenon is also observed in other symbiotic bacteria, but mostly in microorganisms that are directly transmitted from parent to offspring, never having to live on their own. Yet, some evidence suggests that the bacteria in the lure of anglerfish may be spending time in the water until they find a new host, crossing thousands of meters of ocean in the process.

To explore this paradox, Baker et al. looked into the type of bacteria carried by different groups of anglerfish. If each type of fish has its own kind of bacteria, this would suggest that the microorganisms are passed from one generation to the next, and are evolving with their hosts. On the other hand, if the same sort of bacteria can be found in different anglerfish species, this would imply that the bacteria pass from host to host and evolve independently from the fish.

Genetic data analysis showed that amongst six groups of anglerfishes, one species of bacteria is shared across five groups while another is specific to one type of fish. The analyses also revealed that anglerfish and their bacteria are most likely not evolving together. This means that the bacteria must make the difficult journey from host to host by persisting in the deep sea, which was confirmed by finding the genetic information of these bacteria in the water near the fish.

Anglerfish and the bacteria that light up their lure are hard to study, as they live so deep in the ocean. In fact, many symbiotic relationships are equally difficult to investigate. Examining genetic information can help to give an insight into how hosts and bacteria interact across the tree of life.
DOI: https://doi.org/10.7554/eLife.47606.002

pouch-like symbiont-filled protuberances anterior to the dorsal fin, known as caruncles. Bioluminescent symbiosis is thought to be essential to the survival of adult anglerfishes, although the exact function has not been observed. The lure has been proposed to attract prey, confound predators, or signal mates (*Pietsch, 2009*). Recent research by *Hendry et al. (2018)* investigated symbiont genomes from two commonly collected anglerfish species, *Cryptopsaras couesii* and *Melanocetus johnsonii*, host lineages that diverged approximately 100 million years ago (*Miya et al., 2010*; *Pietsch, 2009*). Each host harbored a distinct species of bacterial symbiont: *C. couesii* hosts '*Candidatus* Enterovibrio luxaltus' and *M. johnsonii* hosts '*Candidatus* Enterovibrio escacola,' referred to here as *E. luxaltus* and *E. escacola* for ease (*Hendry et al., 2018*).

Most luminescent bacterial symbionts are facultatively symbiotic, have genome sizes typical of nonsymbiotic, free-living relatives, and are acquired by hosts from environmental populations (*Bongrand et al., 2016*; *Bright and Bulgheresi, 2010*; *Dunlap et al., 2012*; *Dunlap and Urbanczyk, 2013*; *Ruby et al., 2005*; *Urbanczyk et al., 2011*). In contrast, anglerfish symbiont genomes are reduced ~50% relative to closely related free-living bacteria, a pattern more commonly seen in intracellular, obligate symbiosis (*Fisher et al., 2017*; *Kuwahara et al., 2007*; *Manzano-Marín and Latorre, 2016*; *Shigenobu et al., 2000*). Anglerfish symbionts, which have not been successfully cultured (*Haygood et al., 1984*), appear to be obligately dependent on their hosts for growth, as the metabolic capacity to use carbon sources other than glucose are absent from the genome and glucose is an extremely limited resource in the deep sea (*Hansell, 2013*; *Hendry et al., 2018*). Genomic degeneration in obligate symbionts is thought to occur as a result of relaxed purifying selection on genes that are unnecessary within the host habitat (*Bright and Bulgheresi, 2010*; *Kenyon and Sabree, 2014*; *Fisher et al., 2017*; *Sachs et al., 2011*). This process may be mediated in part by relaxed regulation of transposable elements (TEs) (*McCutcheon and Moran, 2011*), which was observed for both *E. escacola* and *E. luxaltus* genomes (*Hendry et al., 2018*). Transposon expansions and pseudogenization were evident in both symbionts, with TE pseudogenes making up about 30% of each bacterial genome (*Hendry et al., 2018*). Phylogenetic investigation of transposon families found

independent expansions within each symbiont species, suggesting that genome reduction may have occurred independently within each lineage (*Hendry et al., 2018*).

Although the aforementioned evolutionary patterns tend to result from physical restriction to hosts and vertical transmission between host generations (*Bright and Bulgheresi, 2010*; *McCutcheon and Moran, 2011*; *Moran, 1996*), several characteristics of anglerfish and their symbionts suggest that these bacteria may be environmentally acquired. The anglerfish symbiont genomes retain genes predicted to be under selection outside the host, such as genes involved in cell wall synthesis and complete motility and chemotaxis pathways. Symbionts are also capable of producing polyhydroxybutyrate (PHB), a carbon storage molecule that is hypothesized to aid in environmental persistence until colonizing suitable hosts (*Haygood, 1993*; *Hendry et al., 2018*; *Hendry et al., 2016*). Furthermore, anglerfish life history traits could preclude the possibility of vertical transmission. Anglerfishes reproduce through the production of an 'egg raft' or 'veil' that delivers eggs to the surface. Juvenile anglerfishes do not have lures; as anglerfishes near sexual maturity they descend to bathypelagic depths and develop their lure (*Pietsch, 2009*). This developmental process and the anglerfish's poor swimming abilities, coupled with differences in surface and deep-sea currents (*Etter and Bower, 2015*; *Pazos and Lumpkin R, 2007*; *Pietsch, 2009*), likely result in generations separated by several kilometers of ocean, making it unlikely that juveniles acquire symbionts from their parents in the deep sea. It also appears unlikely that anglerfishes acquire symbionts from their egg raft, as juveniles have not been found with symbiotic bacteria in their developing lures (*Freed et al., 2019*; *Munk, 1999*).

An alternative hypothesis to vertical transmission is that anglerfishes acquire bacterial symbionts from persistent environmental populations despite the limited metabolic capacity of symbionts. Fishes known to harbor luminous bacteria regularly release symbionts into the environment (*Haygood, 1993*). In ceratioid anglerfishes, the extracellular symbiotic bacteria are likely released from the host via a small opening in the lure (*Haygood et al., 1984*; *Munk, 1999*). Environmental samples of bacteria taken concurrently with anglerfish collections in the Gulf of Mexico found 16S rDNA sequences resembling symbionts (*Freed et al., 2019*), which suggests that anglerfish symbionts could be environmentally acquired. Symbiont transmission between host generations via environmental populations, referred to here as environmental acquisition, is not uncommon in the deep sea; for instance, it occurs in the symbiosis of tubeworms (*Feldman et al., 1997*; *Nussbaumer et al., 2006*) and mussels (*Won et al., 2008*; *Won et al., 2003*) with their chemosynthetic bacteria. However, these bacteria have genome sizes typical of free-living relatives and lack signatures of reduction (*Kleiner et al., 2012*; *Li et al., 2018*; *Ponnudurai et al., 2017*). Marine symbionts with reduced genomes have been found, such as the symbionts of deep-sea clams in the genus *Calyptogena* (*Kuwahara et al., 2007*; *Newton et al., 2007*) or the luminous symbionts of anomalopid flashlight fishes (*Hendry et al., 2016*; *Hendry et al., 2014*). However, these symbionts are characterized as having vertical (*Goffredi et al., 2003*; *Hurtado et al., 2003*) and possibly pseudovertical transmission (*Hendry and Dunlap, 2014*).

Because strictly vertically transmitted symbionts will codiverge with their hosts (*Bright and Bulgheresi, 2010*; *Clark et al., 2000*; *Jousselin et al., 2009*; *Zhang et al., 2018*), we assessed the likelihood of hypothesized transmission modes of anglerfish symbionts by testing for symbiont-host codivergence. This analysis included multiple specimens of the previously studied host species *C. couesii* and *M. johnsonii*, as well as less common genera of anglerfishes, including *Ceratias*, *Chaenophryne*, *Linophryne*, and *Oneirodes*. The geographic distribution of these genera is poorly known, but based on collection data the rarest species in our study (*Linophryne maderensis*), has only four documented museum samples (*Pietsch, 2009*). These host species originate from four of the eleven families of ceratioid anglerfishes and span much of the phylogenetic diversity of the suborder Ceratioidei. We hypothesized that a high degree of congruence between host and symbiont phylogenies will indicate codivergence due to vertical transmission. Additionally, codivergence could result in diverse symbiont species associated with diverse host lineages. Alternatively, if symbiont and host phylogenies lack congruence and different host species share symbionts, this indicates likely acquisition of symbionts from environmental populations.

## Results

### Anglerfish species host only two distinct symbiont species

Contigs closely matching genome sequences of the previously reported luminous symbionts *E. escacola* or *E. luxaltus* were found in all samples. These symbiont species were never recovered together and no additional luminescence genes from other taxa were found in any assemblies, confirming prior findings that individual anglerfish host a single species of symbiont (*Hendry et al., 2018*). Additionally, hosts for which both esca and caruncle samples were available (*Cryptopsaras*) hosted the same symbiont species in both light organs (*Table 1*). Phylogenetic analysis of conserved housekeeping genes confirmed that all new symbiont samples in this study are closely related to previously documented species (*Hendry et al., 2018*) (*Figure 1*). Short or nonexistent branch lengths within each symbiont species clade suggest that there are few genetic differences between samples, which was supported by ANI values (*Table 1*). Within a species there was greater than 99% ANI to the previously identified symbiont species (*Hendry et al., 2018*) and the between symbiont species ANI was less than 74%. All genomes had an average coverage depth of 15x or greater.

### Host-specificity and codivergence

Comparison of the host and symbiont phylogenies showed very little congruence, suggesting that neither symbiont species has co-diverged with their host (*Figure 2*). A symbiont phylogeny was constructed using 205 single-copy protein-coding genes shared by anglerfish symbionts and closely-related free-living bacteria. The construction of a protein-coding phylogeny was employed to get higher-resolution of the relationship between symbionts relative to the house keeping phylogeny. Both analyses showed similar relationships between symbionts (*Figure 2*). This symbiont topology

**Table 1.** Statistics for symbiont genome sequences analyzed in this study.

Samples that are unique to this study are bolded. For binned genomes, the average nucleotide identity (ANI) of the genome compared to the reference sequence is shown. For *E. luxaltus* the reference was the CC26 symbiont and *E. escacola* was the MJ02 symbiont previously documented (*Hendry et al., 2018*). Results indicating similar species using ANI are bolded. Samples that could not be successfully binned and were not included in the ANI and completeness analysis are marked with a '–'. Samples when compared to themselves are marked with 'NA'. Statistics for total length and GC content were generated using OrthoANU, the percent completeness was generated using checkM, and the coverage was generated using BBmap. Sample location is denoted with a ♦ for those from the Northern Atlantic and without notation for those from the Gulf of Mexico.

| Sample | FishID | Light organ | Accession # | E. escacola ANI | E. luxaltus ANI | Length (Mb) | GC content (%) | Complete (%) | Ave coverage |
|---|---|---|---|---|---|---|---|---|---|
| CC26E | *Cryptopsaras couesii* | esca | GCA002300443.1 | 73.7 | NA | 2.14 | 37.7 | 91.3 | 25 |
| CC32E | *Cryptopsaras couesii* | esca | SRR8206628 | – | – | – | – | – | 23 |
| CC81C | *Cryptopsaras couesii* | caruncle | SRR8206630 | – | – | – | – | – | 19 |
| **CCS1E** ♦ | *Cryptopsaras couesii* | esca | RPOE00000000 | 73.6 | **99.9** | 2.14 | 37.7 | 90.8 | 567 |
| **CCS2C**♦ | *Cryptopsaras couesii* | caruncle | RPOF00000000 | 73.7 | **99.9** | 2.20 | 37.6 | 90.3 | 313 |
| CC62E | *Cryptopsaras couesii* | esca | SRR8206629 | – | – | – | – | – | 19 |
| Csp75C | *Ceratias uranoscopus* | caruncle | RPGC00000000 | **99.9** | 73.8 | 2.73 | 39.8 | 91.0 | 1600 |
| CspS10C ♦ | *Ceratias sp.* | caruncle | RPGB00000000 | **99.2** | 73.6 | 2.72 | 39.8 | 91.1 | 99 |
| CspS9C ♦ | *Ceratias sp.* | caruncle | RPGE00000000 | **99.1** | 73.8 | 2.69 | 39.8 | 89.3 | 26 |
| CU44E ♦ | *Ceratias uranoscopus* | esca | RPGD00000000 | **99.1** | 74.0 | 3.04 | 39.8 | 88.3 | 15 |
| CLS4E ♦ | *Chaenophryne longceps* | esca | RPGF00000000 | **99.9** | 73.7 | 2.73 | 39.8 | 90.4 | 330 |
| CDS3E ♦ | *Chaeonophryne sp.* | esca | RPGG00000000 | **99.9** | 73.6 | 2.73 | 39.8 | 89.3 | 291 |
| LMS8E ♦ | *Linophryne maderensis* | esca | RPGH00000000 | **99.8** | 73.8 | 3.40 | 40.0 | 88.8 | 1 |
| MJ02E | *Melanocetus johnsoni* | esca | GCA002381345.1 | NA | 73.7 | 2.65 | 39.8 | 89.9 | 766 |
| MJS5x ♦ | *Melanocetus johnsoni* | esca | RPGI00000000 | **99.9** | 73.8 | 3.09 | 39.8 | 91.1 | 321 |
| DP02E | *Oneirodes sp.* | esca | RPGJ00000000 | **100.0** | 73.7 | 2.68 | 39.8 | 89.3 | 910 |

DOI: https://doi.org/10.7554/eLife.47606.004

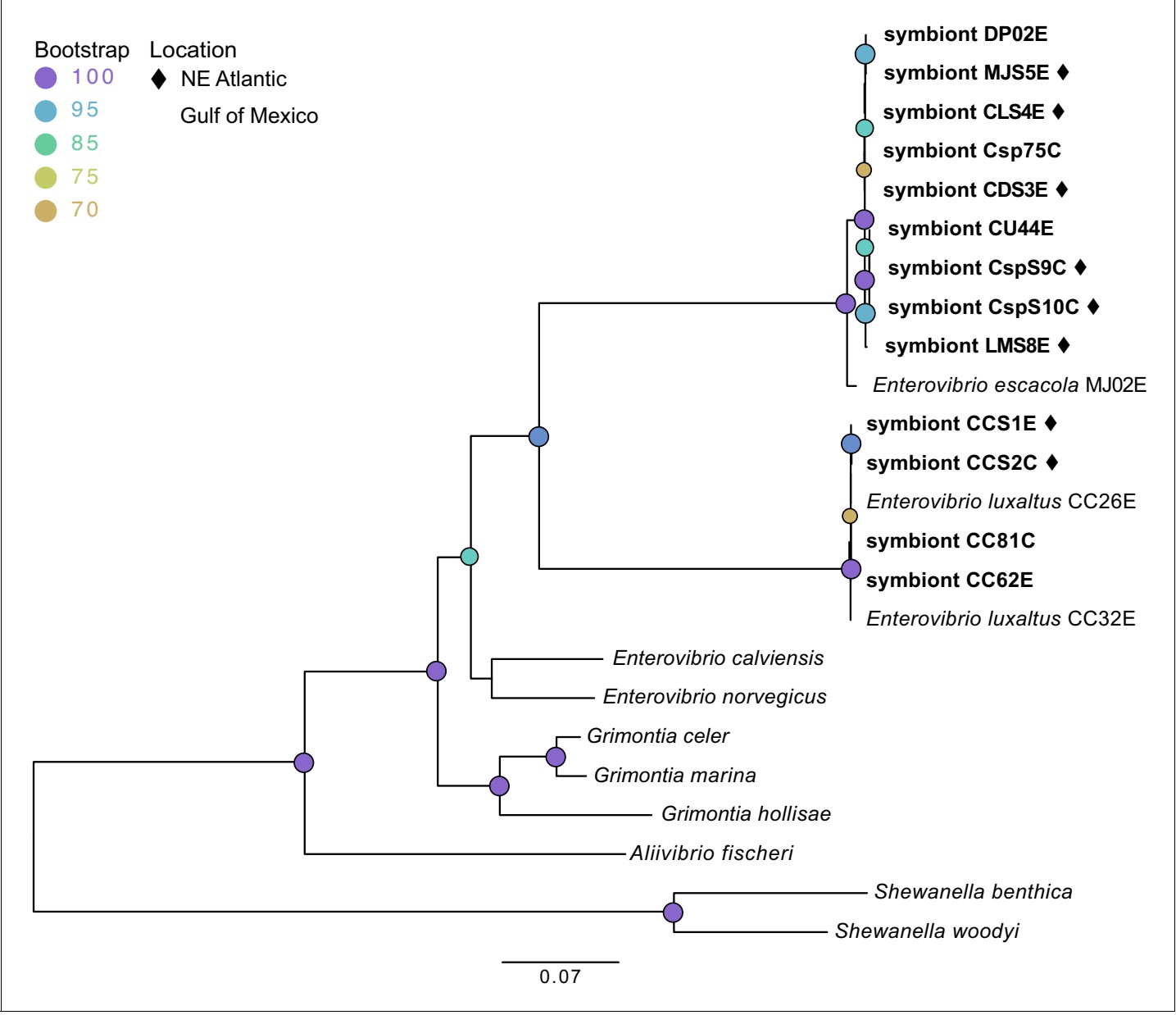

**Figure 1.** Maximum likelihood phylogenetic tree of bacterial symbionts from conserved housekeeping genes: 16S rDNA, *atpA*, *gapA*, *gyrB*, *rpoA*, and *topA*. General time reversible was selected by modelfinder and a tree was constructed using IQ-TREE with 1000 bootstrap replicates. Those samples unique to this study are bolded, with samples from the Northern Atlantic denoted with ♦, and the bootstrap values over 60 are listed at tree nodes. DOI: https://doi.org/10.7554/eLife.47606.003

was compared to a host phylogeny constructed using mitochondrial genes, which matches previous analysis of anglerfish evolutionary relationships (*Miya et al., 2010*) (*Figure 2*). Comparison of host and symbiont phylogenies found *E. luxaltus* was only associated with the fish species *C. couesii*, and that all fish in this clade hosted the same species of symbiont, indicating that *E. luxaltus* and *C. couesii* may have a specific interaction. In contrast, *E. escacola* was the symbiont associated with every other anglerfish sample evaluated. These other fish samples cover much of the diversity in the suborder, including four of the 11 ceratioid families distributed across the phylogeny. These diverse anglerfishes all hosted very genetically similar symbiont lineages that are polyphyletic with respect to host identity.

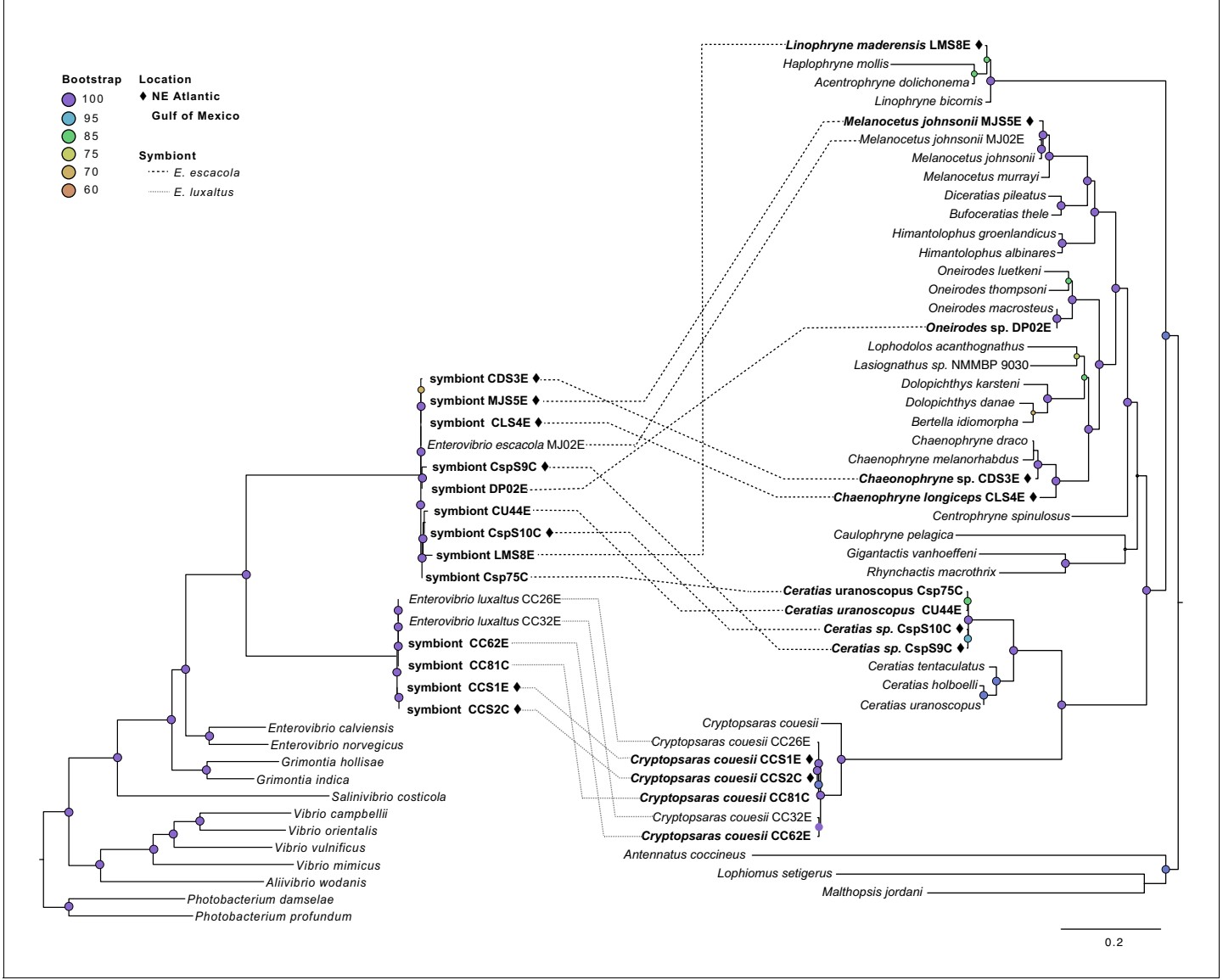

**Figure 2.** Symbiont phylogeny (left) constructed using single-copy protein-coding genes compared to the host phylogeny constructed using mitochondrial genes (right). Bolded samples are unique to this study. Samples from the Northern Atlantic denoted with ♦, and the bootstrap values over 60 are listed at tree nodes. Linkages between symbionts and their hosts are shown with dotted lines that differentiate between symbiont species.
DOI: https://doi.org/10.7554/eLife.47606.005

None of the symbiont phylogenies, including those constructed with single-copy protein-coding genes and conserved housekeeping genes (*Figure 3*), resulted in significant congruence of *E. escacola* and host phylogeny after statistical testing for symbiont-host codivergence using Procrustean superposition of the symbiont phylogeny. This is not surprising as *E. escacola* symbionts from the same genus of host did not form monophyletic groups in any analysis. In the analysis of conserved single-copy protein-coding genes, only two of the Atlantic samples of *E. luxaltus* (symbiont CC32E and CC62E) were significantly congruent with *C. couesii* hosts. Analysis using the housekeeping gene phylogeny resulted in significant congruence with only some host-symbiont pairs, specifically symbionts CC26E, CCS1E, and CCS2E, suggesting that any significant congruence in this clade is not robust. Symbiont and host phylogenies had limited congruence across the anglerfish suborder, and we found no reliable signals of codivergence which might indicate vertical transmission.

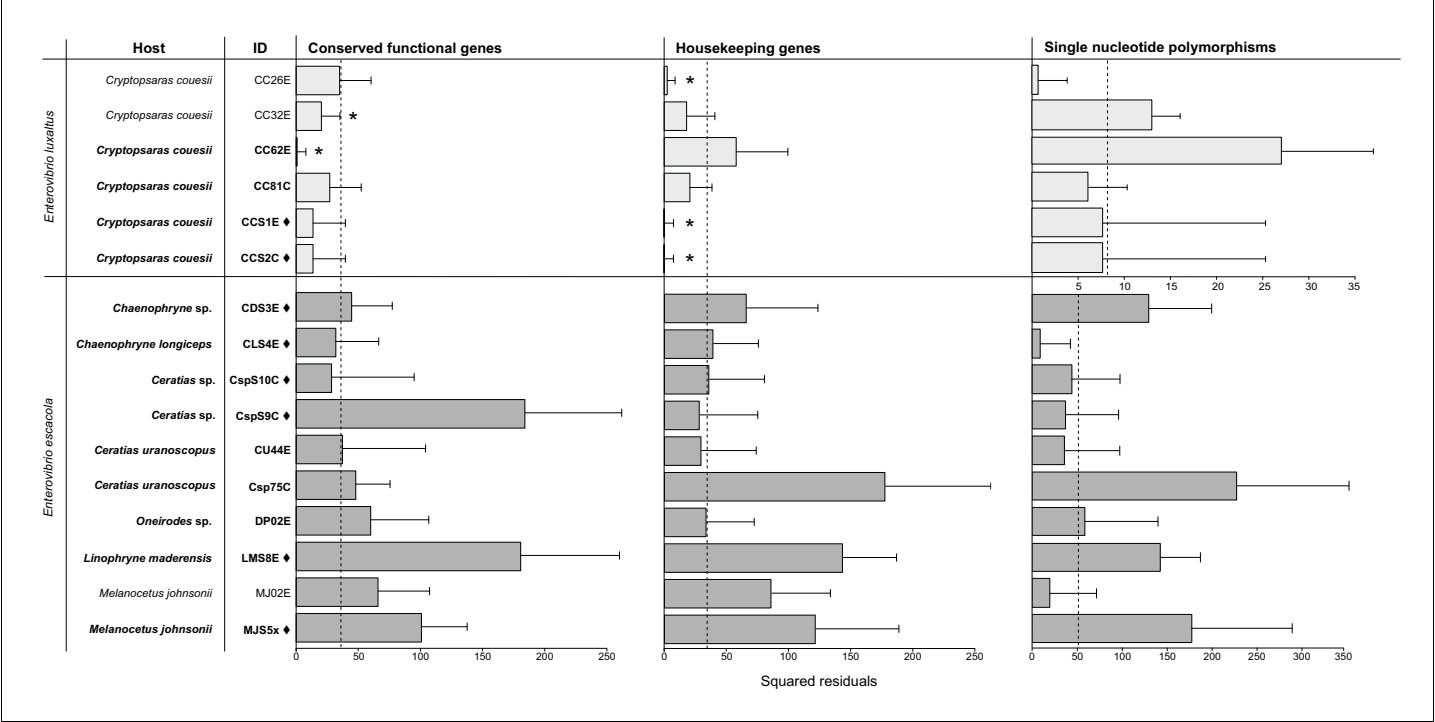

**Figure 3.** Procrustean Approach to Cophylogeny using a host matrix constructed using mitochondrial gene phylogeny compared to symbiont matrices constructed using the single-copy protein-coding gene phylogeny (p=2e-05) and housekeeping genes phylogeny (p=2e-05). SNPs phylogenies were analyzed for each species, and the scale for *E. luxaltus* was dissimilar to the *E. ecacola*; neither were statistically significant (p>0.5 for analysis of both species). The squared residuals below the median squared residual value (dotted line) are significantly codiverging with the host phylogenies (marked with an asterisk). Sample IDs from the Northern Atlantic are marked with a ◆ and those from the Gulf of Mexico are unmarked.

DOI: https://doi.org/10.7554/eLife.47606.006

## Symbiont-specific DNA found in environmental samples

In order to further evaluate the possibility of anglerfish symbionts persisting environmentally, we attempted to amplify DNA from environmental samples with symbiont-specific primers. We used a PCR assay for a highly conserved and species specific 198-basepair portion of the *cheA* locus from each symbiont on seawater bacterial samples. This locus was successfully amplified and sequenced from a subset of the samples, with sequences identified as *E. luxaltus* and *E. escacola* found in distinct environmental samples. Four samples (8% of those evaluated) were confirmed to contain the *E. luxaltus cheA* gene (*Supplementary file 6*). These nucleotide sequences were 99–100% similar to the *cheA* locus in all *E. luxaltus* genomes available (*Hendry et al., 2018*; this study). The amplicon sequences do not appear to be from other known bacteria, such as closely related *Enterovibrio*. The most similar match to the environmental sequences in GenBank databases (non-redundant, Refseq genome, and whole genome shotgun) with >60% coverage shared only 80% nucleotide identity. Four different samples (8% of those evaluated) were identified as *E. escacola*. Sequences of *E. escacola cheA* from the environment did not have significant matches in GenBank databases, but were 99% similar to the *E. escacola* locus from available genomes (*Hendry et al., 2018*; this study). The amplified *cheA* region is only 78% similar between *E. escacola* and *E. luxaltus* and phylogenetic analysis confirmed that the environmental sequences clustered with *E. escacola* and *E. luxaltus* sequences rather than *cheA* orthologs from the highest non-symbiont BLAST matches in GenBank (*Figure 4*). This phylogenetic clustering and high nucleotide identity suggest that the *cheA* locus is highly conserved within each species and distinctive from closely related bacteria, so we conclude that successful amplifications from seawater indicate that the symbionts were present in the environment.

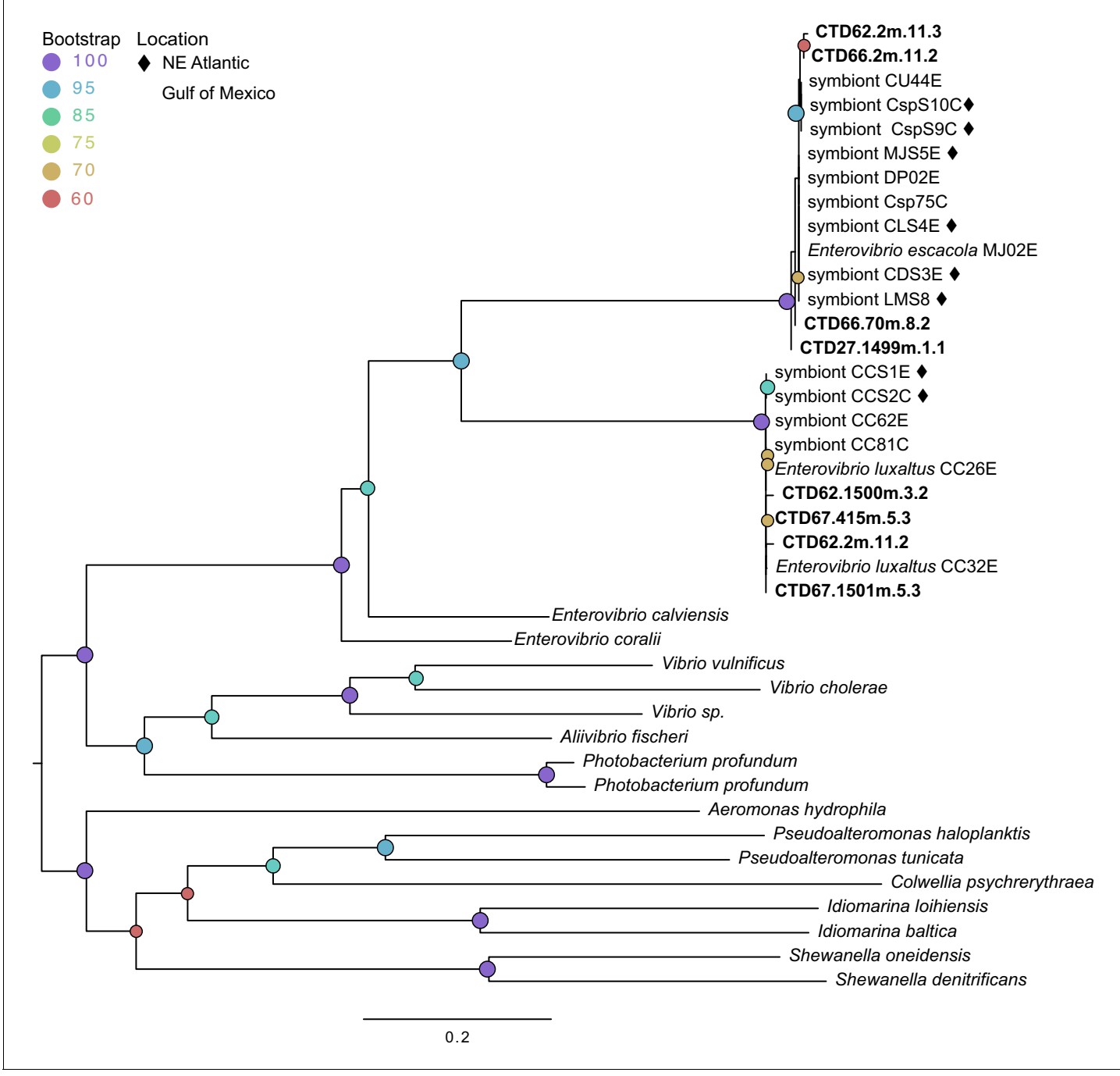

**Figure 4.** Maximum likelihood phylogenetic tree of cheAfrom environmental samples (bolded) compared to sequences from symbiont genomes isolated from fish and sequences from related species. Modelfinder selected the general time reversible model and a tree was constructed using IQ-TREE with 1000 bootstrap replicates. Those samples from the Northern Atlantic denoted with ♦, and the bootstrap values over 60 are given at tree nodes.

DOI: https://doi.org/10.7554/eLife.47606.007

## Within a symbiont species, samples differed by SNPs

There was very little genetic diversity within both *E. escacola* and *E. luxaltus* at the loci analyzed above, which could possibly obscure codivergence between symbionts and hosts. To investigate this possibility, as well as any geographic patterns in symbiont distribution, phylogenies were constructed using more data in the form of genome-wide single nucleotide polymorphisms (SNPs)

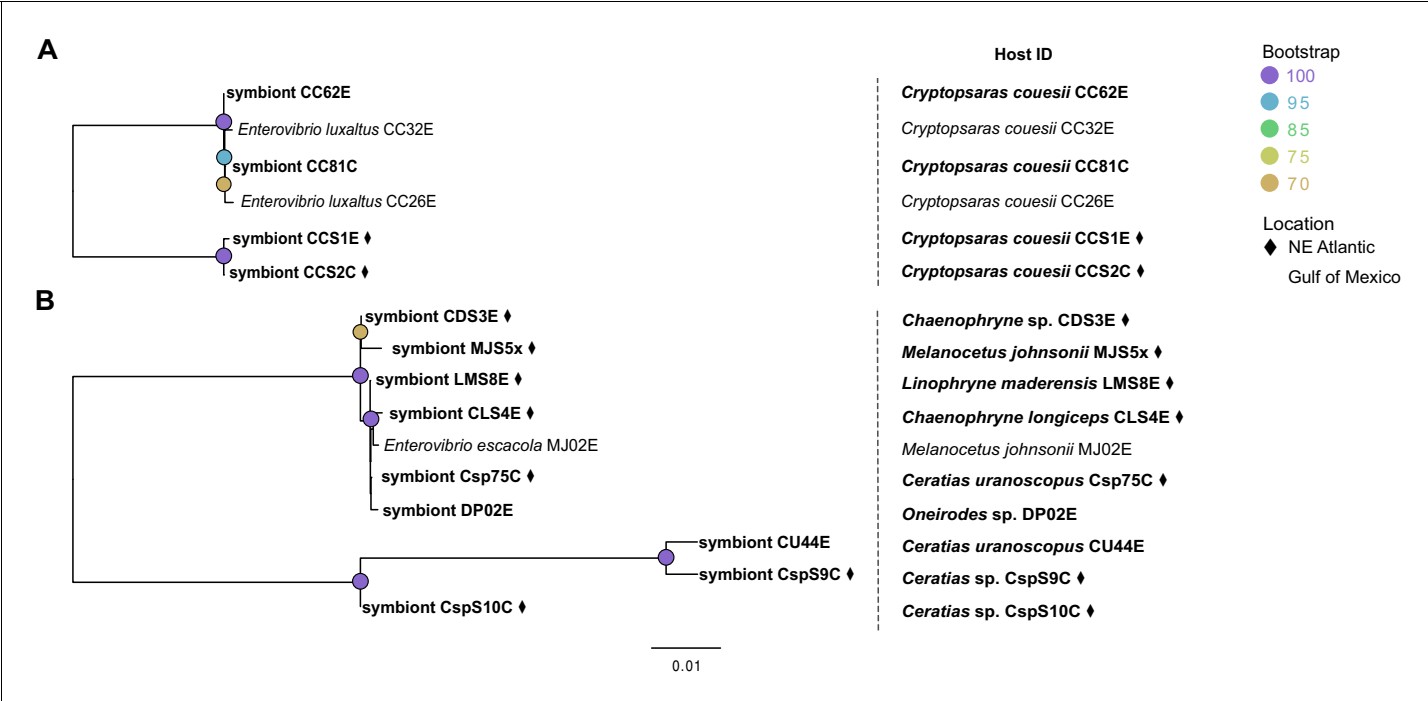

**Figure 5.** Phylogenies constructed using single nucleotide polymorphisms for (**A**) *E. luxaltus* (2252 SNPs) and (**B**) *E. escacola* (15272 SNPs). Host identifications for each sample are listed in the right-hand column. Samples unique to this study are bolded and those from the Northern Atlantic are marked with a ♦.

DOI: https://doi.org/10.7554/eLife.47606.008

(*Figure 5A B*). Fewer SNPs present across samples were found in *E. luxaltus* genomes (2252) compared to *E. escacola* genomes (15272). The *E. luxaltus* SNP phylogeny does further differentiate between samples with high support (*Figure 5A*). Specifically, a host esca (E) and caruncle (C) from the North Atlantic were divergent from those collected in the Gulf of Mexico. However, with the limited number of available samples, we are not able to fully investigate if this is due to geographic patterns or the substantial time between sampling (~20 years) (*Supplementary file 1*). However, samples collected from the same location from different months (with a maximum difference in collection time of 12 months) did not form distinct clades. The SNP phylogeny constructed for *E. escacola* also showed more divergence than phylogenies from conserved genes, but samples did not form distinct clades by location or collection date (*Figure 5B*). Consistent with other analyses, *E. escacola* samples isolated from the same host genera were polyphyletic and there was often greater variation between symbionts isolated from the same anglerfish genus than between symbionts from different host genera. Some *Ceratias* symbiont samples did form a long branch that was distinct from all other *E. escacola* lineages, with the exception of a single *Ceratias* sample (Csp75C). To confirm a lack of codivergence with improved phylogenetic resolution, we performed the Procrustean analysis using the SNP phylogenies, but neither symbiont was significantly codiverging with their hosts (p>0.05) in this analysis.

## Discussion

Within the broad phylogenetic spectrum of ceratioid anglerfishes sampled in this study we identified only two symbiont species, *E. luxaltus* and *E. escacola*. These symbiont species were previously described as the symbionts of two commonly collected anglerfish species, *Cryptopsaras couesii* and *Melanocetus johnsonii*. The fact that sampling from four additional anglerfish genera from two different ocean basins did not uncover more symbiont diversity suggests that there are a low number of luminescent symbiont species that can associate with deep-sea anglerfishes. However, it should be noted that this study was not all-inclusive, particularly since new anglerfish species are still being

described (*Pietsch and Sutton, 2015*), and further sampling could reveal more symbiont diversity. In addition to the lack of species-level diversity, the low intra-specific diversity in each symbiont species was notable. Similar trends have been found in other obligate symbionts, such as the aphid endo-symbiont *Buchnera* and in the obligate luminous symbionts of flashlight fishes (*Hendry et al., 2014*), but not in facultative luminous symbionts (*Abbot and Moran, 2002*; *Funk et al., 2001*; *Hendry et al., 2014*). Host population bottlenecks may lead to low genetic diversity within *Buchnera* (*Abbot and Moran, 2002*; *Funk et al., 2001*), but this seems unlikely to account for low diversity in obligate symbionts of fishes. It is unclear why anglerfish symbionts, even from distinct hosts and geo-graphic locations, are so genetically similar. It is possible that although the hosts are separated by ocean basins, low mutation rates and long doubling times have resulted in a fairly stable and wide-spread symbiont population. Anglerfish symbionts lack many DNA repair pathways (*Hendry et al., 2018*), which has been implicated in increased mutation rates in obligate symbionts (*Lind and Andersson, 2008*), but the connection between the loss of these pathways and genomic evolution is not always clear (*Tamas et al., 2002*). Our finding of low genetic diversity in anglerfish symbionts supports the idea that a loss of DNA repair mechanisms does not necessarily lead to high mutation rates in bacteria. We speculate that anglerfish symbionts may instead have long doubling times, as this adaptation is common for bacteria surviving in the low-nutrient, high-hydrostatic pressure of the deep sea (*Lauro and Bartlett, 2008*; *Wirsen and Molyneaux, 1999*). Lowered metabolic rates are also common for cooperative symbionts (*An et al., 2014*). Collectively these factors may have led to the bacterial genomes being relatively static when free-living and resulted in the limited diversity observed in this study.

Although some obligate bacteria show low genetic diversity within a host species, obligate sym-bionts from different host species are often distinct due to codivergence with their hosts (*Clark et al., 2000*; *Jousselin et al., 2009*). This pattern has been found in numerous symbionts that are known to be vertically transmitted and often results in phylogenetic congruence between distinct host and symbiont taxa (*Fisher et al., 2017*; *Sachs et al., 2011*), as has been documented in the bacterial symbionts of insects (*Dale and Moran, 2006*; *Moran et al., 2008*) and deep-sea clams (*Goffredi et al., 2003*; *Hurtado et al., 2003*). Within vertically transmitted symbionts, symbiont replacements or horizontal transfers can often be observed in specific lineages where congruence breaks down (*Bright and Bulgheresi, 2010*), as has been observed in *Wolbachia*-harboring insects (*Kikuchi and Fukatsu, 2003*; *Lefoulon et al., 2016*), bacterial symbionts of marine worms (*Blazejak et al., 2006*), and Prochloron associated with sea-squirts (*Münchhoff et al., 2007*). Neither of these patterns is seen in our data. Anglerfish symbionts and their hosts lack consistently congru-ent phylogenies and it does not seem likely that congruence is being obscured by symbiont replace-ments or transfers, since very diverse host species all share low diversity symbionts. These results support the hypothesis that anglerfish symbionts are not codiverging with their host species. This conclusion is robust for *E. escacola* and associated hosts, but we may not have enough samples, and genetic diversity within those samples, to rule out the possibility that *E. luxaltus* and *C. couesii* could be codiverging due to vertical transmission. An alternative hypothesis is that *C. couesii* and *E. luxal-tus* have a specific interaction, and that either the host or the bacterium excludes the other symbiont species (*Bongrand and Ruby, 2019*; *Koch et al., 2014*).

A lack of robust and statistically significant codivergence between hosts and symbionts contra-dicts the hypothesis that either symbiont species is vertically transmitted. This is consistent with pre-vious studies of the luminous symbionts of squid and fish hosts, as they show no congruence between host and symbiont species (*Dunlap et al., 2007*). The most likely conclusion based on these data is that anglerfishes acquire their bacteria from an environmental symbiont pool that interacts with diverse anglerfish species. However, anglerfish symbiont genomes resemble vertically transmit-ted symbionts in multiple ways, including having extreme gene loss, expansion of transposable ele-ments, and limited metabolic capacity (*Hendry et al., 2018*). Similar genomic patterns are seen in 'Candidatus Photodesmus' species, the luminous symbiont of anomalopid flashlight fishes (*Hendry et al., 2016*; *Hendry et al., 2014*). Both anglerfish and anomalopid symbionts have evaded culturing efforts and are divergent from known species of luminous bacteria in the Vibrionaceae (*Haygood et al., 1992*; *Haygood and Distel, 1993*; *Hendry et al., 2018*; *Hendry and Dunlap, 2011*). However, anglerfishes do not appear to school, nor do they exhibit diurnal cave dwelling, that is hypothesized to assist in pseudovertical transmission of *Photodesmus* species to flashlight fishes (*Hendry et al., 2016*; *Hendry et al., 2014*). Flashlight fishes and their symbionts lack sufficient

sampling to test for codivergence, but symbiont sequencing from four fish species found high symbiont-host specificity (*Hendry et al., 2014*), a distinct pattern from the results presented here for *E. escacola* and six host species. We are not aware of other bacterial symbionts that have undergone extensive, degenerative genome reduction while maintaining environmental populations and associations with diverse and widespread hosts, as is seen with deep-sea anglerfish symbionts (For an overview of transmission modes and evolutionary patterns in symbionts, see *Table 2*).

Based on catch rates and limited observation, anglerfishes are thought to be relatively solitary, and different life stages are separated by hundreds to thousands of meters of ocean (*Pietsch, 2009*). Anglerfishes are unlikely to encounter environmental symbionts regularly, as symbionts are unlikely to establish widespread populations due to their limited metabolic capabilities (*Hendry et al., 2018*). Other environmentally transmitted luminescent symbionts have much higher host densities to enrich populations in the local environment (*Nealson and Hastings, 1991*). Anglerfishes may have evolved mechanisms to similarly increase the concentration of symbionts in their local environment. A small pore in the lure is likely seeding the environment with symbiotic bacteria (*Munk, 1999*), but the caruncles on *Ceratias* and *Cryptopsaras* species are also a likely source of symbiotic bacteria. The caruncle is not externally luminescent and its function for the fish is not established (*Pietsch, 2009*). Although it is not connected to the esca, the caruncle does connect to the surrounding water through a small distal pore (*Pietsch, 2009*). The conclusion that anglerfishes must acquire their symbionts from potentially sparse environmental populations leads us to propose that the caruncle evolved as a mechanism to increase the concentration of symbiotic bacteria in the environment, thereby increasing the likelihood of symbionts being transmitted to new fish generations.

**Table 2.** A summary of modes of symbiont transmission, examples of some bacterial species and the functions they perform for animal hosts, and trends in the reduction of symbiont genomes.

| Transmission | Description | Symbiont and function | Host | Genome | References |
|---|---|---|---|---|---|
| Environmental | Acquired from free-living bacteria  | Luminescence<br>*Aliivibrio fischeri*<br>*Photobacterium leiognathi*<br>*Photobacterium kishitanii*<br>Nutrition<br>"*Candidatus* Endoriftia persephone"<br>Various Gammaproteobacteria<br>*Burkholderia* spp. | Fish and squid<br>Fish<br>Fish<br><br>Tubeworms<br>Mussels<br>Insects | Comparable to free-living relatives  | *Dunlap and Urbanczyk, 2013*;<br>*Gyllborg et al., 2012*<br>*Urbanczyk et al., 2011*; *Ast et al., 2007*<br>*Li et al., 2018*; *Kleiner et al., 2012*<br>*Ponnudurai et al., 2017*<br>*Kikuchi et al., 2005*; *Kikuchi et al., 2007* |
| Proposed Environmental | Environmentally persistant cells  | Luminescence<br>"*Candidatus* Enterovibrio escacola"<br>"*Candidatus* Enterovibrio luxaltus" | Anglerfish<br>Anglerfish | Ongoing reduction | *Hendry et al., 2018*<br>*Hendry et al., 2018* |
| Mixed | Pseudovertical or surface transmission  | Luminescence<br>"*Candidatus* Photodesmus blepharus"<br>"*Candidatus* Photodesmus katoptron"<br>Nutrition<br>Various Gammaproteobacteria<br>"*Candidatus* Ishikawaella capsulata" | Flashlight fish<br>Flashlight fish<br><br>Clams<br>Stink bug | Moderate to extreme reduction  | *Hendry et al., 2014*; *Hendry and Dunlap, 2014*<br>*Hendry et al., 2014*; *Hendry and Dunlap, 2014*<br><br>*Kuwahara et al., 2007* |
| Inherited | Direct passage from parent to offspring on egg or sperm  | Nutrition<br>*Buchnera aphidicola*<br>*Carsonella ruddii*<br>*Portiera aleyrodidarum*<br>Varied<br>"*Candidatus* Synechococcus spongiarum" | Aphids<br>Psyllids<br>Whiteflies<br><br>Sponges | Greatly reduced  | *Moran et al., 2008*; *Fisher et al., 2017*<br>*Moran et al., 2008*; *Fisher et al., 2017*<br>*Moran et al., 2008*; *Fisher et al., 2017*<br><br>*Gao et al., 2014*; *Burgsdorf et al., 2015* |

DOI: https://doi.org/10.7554/eLife.47606.009

Low host densities could also drive bacterial evolution in this system. Although *M. johnsonii* is relatively abundant among anglerfishes, as is *C. couesii*, all other anglerfish genera investigated in this study are less prevalent than the most common species (*Pietsch, 2009*). With extremely low host densities, *E. escacola* may remain a viable symbiont for diverse anglerfish species due to selection against host specificity. The lack of apparent host specificity in this symbiont may increase the likelihood that these bacteria in the environment will encounter a permissive host before losing viability. In the case of *E. luxaltus*, this symbiont may encounter sufficient abundances of *C. couesii* individuals to allow for host specificity, either as a result of selection or by chance. For example, *E. luxaltus* and *E. escacola* differ in the genes present in the structural symbiosis polysaccharide (syp) pathway, the regulation of which influences host specificity in the luminous symbiont *A. fischeri* (*Mandel et al., 2009*). Both *sypF* and *sypG* are exclusive to *E. luxaltus*, that is neither gene is present in annotations and cannot be found in a BLAST search of any *E. escacola* genome. In *A. fischeri*, SypF is predicted to be a sensor kinase that regulates biofilm formation (*Darnell et al., 2008*; *Thompson et al., 2018*) and SypG is a response regulator that directly activates the syp locus (*Hussa et al., 2008*; *Thompson et al., 2018*; *Yip et al., 2005*). Although both symbionts maintain genes in the syp pathway, the loss of these regulatory genes in *E. escacola* could facilitate their colonization of a greater diversity of hosts.

The finding that anglerfish symbionts are likely environmentally transmitted further supports the hypothesis that the limited functional capacity of anglerfish luminous symbionts is sufficient to persist in the deep sea before contacting a new host. Motile symbionts capable of chemotaxis may be able to out-compete other non-motile deep-sea bacteria for access to the high nutrient environment of the host esca (*DeLong et al., 2006*). Additionally, polyhydroxybutyrate (PHB) may be a sufficient carbon source to sustain the symbiont before it arrives at a new host (*Hendry et al., 2018*). PHB has been estimated to sustain rhizobia for years, and may assist these microbes to survive thousands of years in a dormant state (*Johnson et al., 2007*; *Muller and Denison, 2018*); we hypothesize that PHB should function similarly for flashlight fish and anglerfish symbionts. Environmental samples of free-living bacteria collected by the DEEPEND Consortium taken concurrently with anglerfish collection found multiple samples containing 16S rDNA matching anglerfish symbionts, at various depths, and from multiple sampling efforts (*Freed et al., 2019*). Our characterization of symbionts using a portion of the chemotaxis protein *cheA* found multiple environmental samples containing either *E. luxaltus* or *E. escacola*. This result confirms previous reports that the symbiont persists in the water column (*Freed et al., 2019*), and further supports our conclusion that symbionts are acquired from environmental populations.

The low genetic diversity within the anglerfish symbionts made it difficult to determine if symbiont distribution was impacted by geographic origin or host identity. Using SNPs we were able to discern differences between *E. luxaltus* samples collected from different times at different locations, however, this result was limited to a single *C. couesii* individual from the North Atlantic, with esca and caruncle sampled (symbiont CCS1E and CCS2C). A subset of the *E. escacola* hosted by *Ceratias* formed a distinct clade and included two of the Northern Atlantic samples and a sample from the Gulf of Mexico. This confounding result suggests that further sampling of *Ceratias* may provide more insight into how location and time impact the diversity of *E. escacola*. Alternatively, it is possible that because the two collection sites are connected by deep-sea currents (Loop Current/Gulf Stream system), that symbionts were acquired by fishes in a similar location although they were collected hundreds of kilometers apart.

The deep sea is the earth's largest and most understudied ecosystem, where studying symbiosis is both challenging and costly. In this study we use genomic analysis on rare samples of one of the deep sea's most prominent symbioses to answer an outstanding question, how are deep-sea anglerfish symbionts transmitted between generations? Our findings demonstrate the value of studying relatively rare organisms in this ecosystem, as we can uncover new findings that may contrast with model systems. Bioluminescent symbiosis in anglerfishes breaks with several expectations from well-studied symbioses; symbionts that leave the host and establish environmental populations typically do not undergo genome degeneration. Yet, here we show that a luminous bacterial symbiont with an extremely reduced genome is able to traverse the low-nutrient, high-pressure environment of the deep sea to establish a symbiosis with a dispersed and relatively rare host. As samples of these fishes and symbionts become available, we may be able to address additional outstanding

questions, such as the lack of diversity in anglerfish symbionts and their biogeographic population structure.

## Materials and methods

### Genome sequencing, assembly, and annotation

Anglerfish samples were collected in the Gulf of Mexico by the DEEPEND Consortium and from east of the Cape Verde Islands by Spencer Nyholm and Peter Herring on the RRS *Discovery* expedition D243 (sample information in *Supplementary file 1*). Morphological identification was done on ship by Tracey Sutton (DEEPEND) or Spencer Nyholm and Peter Herring. Molecular genetic confirmation of morphological identification is discussed below. Samples were named according to: initial morphological identification, order collected, and anglerfish light organ sampled–either esca (E) or caruncles (C). Lures were collected immediately after identification using a sterile scalpel and stored in ethanol or RNAlater (Qiagen, Hilden, Germany) at −80°C until processing. DNA extraction from samples collected in the Gulf of Mexico was performed at the Marine Microbiology and Genetics Laboratory at Nova Southeastern University's Halmos College of Natural Sciences and Oceanography using the PowerLyzer PowerSoil kit (MoBio) as is described in *Hendry et al. (2018)* Samples collected in the Northern Atlantic were extracted using the DNeasy Blood and Tissue Kit (Qiagen). Paired-end 250 base pair Illumina sequence libraries were prepared using the Nextera kit (Illumina, San Diego, CA) and sequenced using HiSeq2500 at the Cornell University Institute of Biotechnology Resource Center Genomics Facility. Contigs were assembled using DISCOVAR de novo and binned and assessed for quality using multiple approaches which are detailed in the Supplementary Information. Binned symbiont genomes and sequences mapped to the reference genomes for *E. luxaltus* and *E. escacola* (GCA_002381345.1 and GCA_002300443.1) were submitted to NCBI (*Supplementary file 1*).

### Genome assembly and validation

High concentrations of an evident monoculture of symbionts within anglerfish escae enable assembly and study of symbiont genomes from samples that are technically metagenomic, as they include symbiont and host DNA as well as DNA from contaminant bacteria likely on the surface of the light organ (*Hendry et al., 2018*). After assembly using DISCOVAR de novo, bacterial genomes were binned using *metabat2*, which bins similar contigs according to tetranucleotide frequency and sequencing depth (*Kang et al., 2015*). Sequences that failed to bin using *metabat2* were binned using the Pathosystems Resource Integration Center (PATRIC 3.5.23) (*Wattam et al., 2014*). Three *C. couesii*-associated samples were not successfully binned using metabat2 or PATRIC; these sample assemblies were processed as is outlined, with the exception of finding average nucleotide identity or annotating gene content. Binned contigs were evaluated through a local BLAST search for genes within the luciferase operon (*luxA*, *luxB*, and *luxC*) and contigs in the resulting bin were input into the NCBI BLAST database to confirm symbiont identification. The average genome coverage depth was calculated using BBmap (*Bushnell, 2014*). Genome completeness was evaluated using checkM (*Parks et al., 2015*), which previously estimated for *E. luxaltus* and *E. escacola* as only nearing 90% completion due to genome reduction (*Hendry et al., 2018*). The quality of the genome assemblies unique to this study are similar to previously documented anglerfish symbionts. Contig bins which had approximately 90% genomic completion, *lux* luminescence genes, and high BLAST similarity to previously sequenced anglerfish symbiont genomes were consider complete symbiont genome sequences and were submitted to Rapid Annotation using Subsystem Technology (RAST) for annotation. All other bins generated by metabat2 and PATRIC did not contain luciferase genes nor did they have sequences that closely resembled symbiont housekeeping genes.

### Anglerfish host evolution

Anglerfish morphological identification and evolutionary relationships among samples were evaluated using mitochondrial genes. Similar methods and comparison species are discussed in *Miya et al. (2010)*. Anglerfish mitochondrial sequences were identified using a local BLAST search of the unbinned contigs and deposited in GenBank (accession numbers MK118159-MK118174). Reference mitochondrial sequences were selected based on initial morphological identifications and

supplemented with sequences of nearest neighbors present in GenBank and *Miya et al., 2010* (*Supplementary file 2*). Mitochondrial sequences were aligned using MAFFT and a phylogenetic tree was assembled using IQ-TREE (*Katoh et al., 2002*; *Nguyen et al., 2015*). Within IQ-TREE modelfinder selects a phylogenetic model using a model-selection method that concurrently searches model and tree space to increase the accuracy of phylogenetic estimates (*Kalyaanamoorthy et al., 2017*). A consensus tree was constructed using the general time reversible model with empirical base frequencies, allowing for invariable sites, and four rate categories (GTR+F+I+G4) and 1000 bootstrap replicates. Based on phylogenetic analysis, samples were assigned to a species if they fell within the same clade as multiple representatives of the same species or by morphological species identification if sequences from representative species were not available for comparison. Samples were identified to a genus if there was an indeterminate species designation. The genetic identification of a single sample contradicted morphological identification, was reevaluated morphologically, and found to confirm the mitochondrial identification.

## Evaluation of symbiont genomes

Similarity among symbiont genomes isolated from individual anglerfish samples was evaluated using average nucleotide identity (ANI), housekeeping genes, and conserved single-copy protein-coding genes. ANI, a measure of nucleotide-level genomic similarity, was found using orthoANIu (*Yoon et al., 2017*); comparisons greater than 95% ANI considered the same species (*Konstantinidis and Tiedje, 2005*). Bacterial species trees were created using conserved housekeeping genes (16S rRNA gene, *atpA*, *gapA*, *gyrB*, *pyrH*, *rpoA*, *topA*) from both symbiont contigs and closely related bacterial genomes downloaded from NCBI (*Supplementary file 3*). Genes were aligned using MAFFT and a tree was constructed from the concatenated alignments in IQ-TREE as described above (GTR+F+I+G4 with 1000 bootstrap replicates). Single-copy protein-coding genes shared by bacterial symbionts and whole genome sequences of closely related free-living bacteria (*Supplementary file 4*) were found by inputting RAST protein annotations into PhyloPhlAn. DNA sequence of shared proteins were then extracted from RAST annotations and used to construct a phylogenomic tree by aligning individual genes in MAFFT. The 205 shared genes were concatenated, and a tree was constructed from 331103 positions using the GTR+F+I+G4 model selected by modelfinder and using 1000 bootstrap replicates in IQ-TREE.

## Evaluating codivergence between anglerfish and bacterial symbionts

Host-symbiont codivergence was evaluated using Procrustean Approach to Cophylogeny (PACo) as implemented in R (*Balbuena et al., 2013*; *R Development Core Team, 2012*). PACo is a global fit method that does not require fully resolved phylogenies to evaluate if the symbiont has evolved as a result of codivergence with the host species. In PACo, Procrustes superposition manipulates the symbiont genetic distance matrix to fit the host matrix, to evaluate the congruence of the symbiont to the host tree. Anglerfish phylogenies input into PACo were constructed as described above for mitochondrial sequences. Various bacterial phylogenies were analyzed in PACo, including the conserved housekeeping gene phylogeny, genome-wide SNP phylogenies (described below), and the conserved single-copy protein-coding gene (identified by PhyloPhlAn) phylogeny (*Segata et al., 2013*). Symbiont and bacterial ultrametic trees were input into PACo as distance matrices, and $10^4$ iterations were performed for significance testing. The contribution of each bacterial symbiont to the overall global codivergence was evaluated using jackknife estimation of the relative squared residuals; codivergence was indicated in those samples that have a significantly smaller fraction of the sum of squares.

## Evaluating presence/absence of bacterial symbiont DNA in water samples

The 16S rDNA sequences matching anglerfish symbionts were previously found in environmental samples taken concurrently with anglerfish collections (*Freed et al., 2019*), suggesting that symbionts persist outside the host. To confirm that anglerfish symbionts can persist in the environment, symbiont species-specific primers were developed from whole genomes to amplify multicopy loci of a conserved chemotaxis protein *cheA*, which should be relatively more abundant than single copy loci in low density samples. We performed PCR assays on DNA extracted from environmental

bacteria in 52 samples taken concurrently with anglerfish collections during DEEPEND consortium cruises (D01-D04) in the Gulf of Mexico. The filtering and extraction protocol used, as well as the 16S rDNA composition of a subset of these samples is described in Easson et al. (*Easson and Lopez, 2019*). Primers for *cheA* specific to each symbiont were designed by importing symbiont and closely related sequences found using the BLAST genome searches into DECIPHER (*Wright et al., 2012*) (*Supplementary file 5*). Sequences were amplified using nested PCR primers and the New England Biolabs standard taq polymerase kit (NE Biolabs, Ipswich, MA, USA) using the recommended protocol for amplifications under 500 base pairs. Reactions were prepared in a UV sterilized biosafety cabinet with surface sterilized implements. Negative controls prepared with sterile water were included in each round of PCR. No negative controls resulted in visible amplification. Amplifications were gel extracted using the Qiaquick gel extraction kit (Qiagen, Venlo, Netherlands) and Sanger Sequenced (Genewiz, New Jersey, USA). Sequence identity was evaluated using blastx and blastn searching and a phylogenetic tree was constructed using MAFFT and IQ-TREE (GTR+F+I+G4 and 1000 bootstrap replicates) from environmental amplifications and *cheA* sequences annotated from genomes available in RAST (*Supplementary file 6*).

## Evaluation of symbiont diversity within a species using SNPs

Evolution within each symbiont species was evaluated using genome-wide SNPs. Bacteria were grouped into different species based on the result of ANI and the conserved housekeeping gene phylogeny. SNPs were identified using snippy v.4.0-dev, which implements bwa mem and freebayes to compare reads from haploid genomes to a reference genome (*Garrison and Marth, 2012*; *Li, 2013*; *R Development Core Team, 2012*; *Seemann, 2015*). The reference genome was selected from the previously characterized anglerfish symbionts described in *Hendry et al. (2018)*. SNPs were identified in sequence reads and snippy-core was used to generate a core alignment of SNPs common to all samples. A phylogenetic tree was constructed using this core alignment in IQ-TREE for each symbiont species with 1000 bootstrap replicates, with the models selected for by modelfinder. The *E. luxaltus* SNPs phylogeny was constructed using the Kimura 3-parameter and ascertainment bias correction model (K3P+ASC) and the *E. escacola* SNPs phylogeny was constructed using the symmetric model with unequal rates and an ascertainment bias correction model (SYM+ASC).

## Acknowledgements

We thank Peter Herring for identification of samples and the crew of the *RRS Discovery* who were instrumental in collections during cruise D243 in the Northeast Atlantic. We thank the DEEPEND 'Deep Pelagic Nekton Dynamics of the Gulf of Mexico' Consortium for the collection of the Gulf of Mexico fish specimens. This research was made possible in part by a grant from The Gulf of Mexico Research Initiative. All data are publicly available through NCBI and the Gulf of Mexico Research Initiative Information and Data Cooperative (GRIIDC) at https://data.gulfresearchinitiative.org (doi: 10.7266/N7P55KWX, 10.7266/N7VX0DK2, 10.7266.N7R49NTN, 10.7266.N70P0X3T, 10.7266/N7XP7385, and 10.7266/N7902234) .

We also thank the Cornell Genomics Facility for consultation and assistance with genomic sequencing.

## Additional information

### Funding

| Funder | Author |
| --- | --- |
| Gulf of Mexico Research Initiative | Jose V Lopez<br>Lindsay L Freed<br>Cole G Easson<br>Tracey T Sutton<br>Danté Fenolio |

The funders had no role in study design, data collection and interpretation, or the decision to submit the work for publication.

## Author contributions
Lydia J Baker, Conceptualization, Data curation, Formal analysis, Investigation, Visualization, Methodology, Writing—original draft, Writing—review and editing; Lindsay L Freed, Cole G Easson, Danté Fenolio, Tracey T Sutton, Spencer V Nyholm, Resources, Investigation, Writing—review and editing; Jose V Lopez, Resources, Funding acquisition, Investigation, Writing—review and editing; Tory A Hendry, Conceptualization, Resources, Data curation, Supervision, Funding acquisition, Investigation, Methodology, Writing—original draft, Project administration, Writing—review and editing

## Author ORCIDs
Lydia J Baker ⬚ https://orcid.org/0000-0002-1453-421X
Jose V Lopez ⬚ https://orcid.org/0000-0002-1637-4125
Tory A Hendry ⬚ https://orcid.org/0000-0002-8001-1783

## Decision letter and Author response
Decision letter https://doi.org/10.7554/eLife.47606.022
Author response https://doi.org/10.7554/eLife.47606.023

# Additional files

## Supplementary files
• Supplementary file 1. Collection data for anglerfish samples used in this study. Gulf of Mexico samples were collected by the DEEPEND consortium at the given locations. Samples collected by S. Nyholm were collected by RRS Discovery expedition D243 east of the Cape Verde Islands at the given locations. A single sample (DP02E) was given a new taxonomic identification as a result of mitochondrial gene analysis.
DOI: https://doi.org/10.7554/eLife.47606.010

• Supplementary file 2. Lophiiform mitochondrial sequences used in this study. Genbank accession numbers and total sequence length are shown. Samples listed in bold were generated by this study or by *Hendry et al. (2018)*.
DOI: https://doi.org/10.7554/eLife.47606.011

• Supplementary file 3. Sequences from free-living Vibrionaceae species used to generate the housekeeping gene tree. Genbank accession numbers of each gene are given.
DOI: https://doi.org/10.7554/eLife.47606.012

• Supplementary file 4. Genomes from free-living Vibrionaceae species used for PhyloPhlan analysis. Genbank accession numbers of each genome assembly are given.
DOI: https://doi.org/10.7554/eLife.47606.013

• Supplementary file 5. The species-specific primers designed to amplify the chemotaxis protein cheA.
DOI: https://doi.org/10.7554/eLife.47606.014

• Supplementary file 6. Species-specific primers designed to amplify the chemotaxis protein *cheA*.
DOI: https://doi.org/10.7554/eLife.47606.015

• Transparent reporting form
DOI: https://doi.org/10.7554/eLife.47606.016

## Data availability
The data used to generate figures can be found in the supplementary material, denoted by "S" in text. The symbiont genomes have been submitted to the SRA database and their accessions have been listed in Table 2 and Supplementary file 1. The fish mitochondrial sequences used in this study are listed in Supplementary file 2. Mitochondrial sequences generated in this study can be found in Genbank (MK118159-MK118174). The Vibrionaceae species used to generate Figure 1 are listed in Supplementary file 3. The Vibrionaceae species used to generate the conserved single-copy protein-coding genes is listed in Supplementary file 4. The bacterial species used to generate Figure 4 can be found in Supplementary file 6. The sequences produced as a result of this study are also listed

and are in Genbank (MK457129-MK457136). All methods are outlined and all statistical analysis are described in detail in the manuscript such that they can be repeated.

The following datasets were generated:

| Author(s) | Year | Dataset title | Dataset URL | Database and Identifier |
|---|---|---|---|---|
| Tory A Hendry, Lydia J Baker | 2018 | Candidatus Enterovibrio luxaltus Genome sequencing and assembly | https://www.ncbi.nlm.nih.gov/bioproject/501852 | NCBI BioProject, PRJNA501852 |
| Tory A Hendry, Lydia J Baker | 2018 | Diversity of anglerfish luminescent symbionts E. escacola | https://www.ncbi.nlm.nih.gov/bioproject/501851 | NCBI BioProject, PRJNA501852 |

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
