## [Decision Letter]

Thank you for submitting your article "Diverse deep-sea anglerfishes share a genetically reduced luminous symbiont that is acquired from the environment" for consideration by *eLife*. Your article has been reviewed by two peer reviewers, and the evaluation has been overseen by Patricia Wittkopp as the Senior and Reviewing Editor. The reviewers have opted to remain anonymous.

The reviewers have discussed the reviews with one another and the Reviewing Editor has drafted this decision to help you prepare a revised submission.

Summary:

This study provides new insights into the process of symbiont transmission for ceratioid anglerfishes, a challenging group of fish to study due to their unique life histories and deep water habitats. Applying NGS techniques, the authors were able to identify the luminous symbionts of these fish sampled from various locations and from several different host species. Importantly, this study demonstrates the ability of such tools to make non-model systems more accessible. The results reveal an interesting pattern where one species of luminous bacteria associates with several different hosts, whereas the other related symbiont seems to only associate with a single host species. The authors ex-lain well why they think that these two patterns were observed and they even suggest a molecular pathway that could be involved in host specificity. Overall, the study is well-done and the authors have considered the possibility that they may have under sampled anglerfishes and might be missing some symbiont genetic diversity.

Essential revisions:

Subsection “Symbiont-specific DNA found in environmental samples”: Please add whether the 198 bp locus in cheA that was used to detect the symbionts in seawater was well-conserved or highly variable (or something else).

---

## [Author Response]

Essential revisions:Subsection “Symbiont-specific DNA found in environmental samples”: Please add whether the 198 bp locus in cheA that was used to detect the symbionts in seawater was well-conserved or highly variable (or something else).

This is an important point and we took several measures to ensure that, as much as possible, we were amplifying a symbiont specific marker while not excluding symbionts that might vary in sequence. We targeted the *cheA* region because it is very conserved within each symbiont species, with greater than >99% nucleotide identity among the 10 *E. escacola* sequences available and the 6 *E. luxaltus* sequences available. However, this region is only 78% similar when sequences from the two symbiont species are compared to each other. This pattern was confirmed with the addition of 8 unique sequences from the water samples, which show >99% similarity to the sequences extracted from genomes. We conclude that the *cheA* locus is highly conserved in each symbiont species and therefore that sequences amplified from the water are unlikely to come from other organisms (see phylogenetic tree in Figure 1). This information was more explicitly stated in the revised manuscript in the Discussion.

To confirm that our primers would not amplify non-target organisms from the environmental samples, we BLASTed this locus against the GenBank nr, reference genomes, and wgs databases. Either no significantly similar sequences were found (in the case of *E. escacola*) or the top hits had <81% sequence identity (*E. luxaltus*). We cannot rule out the possibility that sequences from other microbes in the environment could be amplified with our methods, but we think that non-target species are unlikely to contribute to the pattern we show. We have clarified this in the Discussion.